# Recursive Monte Carlo and Variational Inference with Auxiliary Variables

Alexander K. Lew[1]                Marco Cusumano-Towner[1]                Vikash K. Mansinghka[1]

[1]Massachusetts Institute of Technology, Cambridge, Massachusetts, USA

## Abstract

A key design constraint when implementing Monte Carlo and variational inference algorithms is that it must be possible to cheaply and exactly evaluate the marginal densities of proposal distributions and variational families. This takes many interesting proposals off the table, such as those based on involved simulations or stochastic optimization. This paper broadens the design space, by presenting a framework for applying Monte Carlo and variational inference algorithms when proposal densities cannot be exactly evaluated. Our framework, *recursive auxiliary-variable inference* (RAVI), instead approximates the necessary densities using *meta-inference*: an additional layer of Monte Carlo or variational inference, that targets the proposal, rather than the model. RAVI generalizes and unifies several existing methods for inference with expressive approximating families, which we show correspond to specific choices of meta-inference algorithm, and provides new theory for analyzing their bias and variance. We illustrate RAVI's design framework and theorems by using them to analyze and improve upon Salimans et al. [35]'s Markov Chain Variational Inference, and to design a novel sampler for Dirichlet process mixtures, achieving state-of-the-art results on a standard benchmark dataset from astronomy and on a challenging data-cleaning task with Medicare hospital data.

## 1 INTRODUCTION

Monte Carlo and variational inference algorithms are the workhorses of modern probabilistic inference, a fundamental problem with applications in many disciplines [25]. A key challenge in applying these algorithms is the design of *proposal distributions* (in VI, variational families), which

can greatly affect their performance [5]. A good proposal should incorporate any knowledge the practitioner might have about the shape of the posterior; however, this goal is often in tension with the requirement that a proposal's marginal density be analytically tractable, in order to compute importance weights, MCMC acceptance probabilities, or gradient updates for VI. The challenge is that proposal distributions that are simple enough to admit exact density evaluators may not be flexible enough to solve real-world posterior inference problems.

In this paper, we present a new framework, called *Recursive Auxiliary-Variable Inference* (RAVI), for incorporating more complex proposals, without exact marginal density evaluators, into standard Monte Carlo and VI algorithms. The key idea is to approximate the proposal densities using *meta-inference* [8]: an additional layer of Monte Carlo or variational inference targeting the proposal, rather than the model. RAVI generalizes and unifies several existing methods for inference with expressive proposals [35, 33, 38], which we show correspond to specific choices of meta-inference algorithm (see Appendix B for 10 examples).

**Contributions.** Our key contributions are:

- the RAVI framework, including new recursive algorithms for IS, VI, SMC, and MH using proposals without exact marginal density evaluators (Sections 2 & 3);
- theorems characterizing the impact of RAVI's estimated densities on inference quality (sampler variance, or tightness of variational bounds) (Section 4); and
- two extended examples of RAVI's application to algorithm design and analysis: (1) a novel variant of Salimans et al. [35]'s Markov Chain Variational Inference (MCVI) algorithm that, unlike vanilla MCVI, scales to handle proposals incorporating long MCMC chains; and (2) a novel sampler for Dirichlet process mixtures that uses a randomized agglomerative clustering algorithm as a proposal, outperforming strong baselines on a standard benchmark from astronomy [14] and a challenging Medicare data cleaning problem [23, 19].

*Accepted for the 38th Conference on Uncertainty in Artificial Intelligence* (UAI 2022).

| Monte Carlo or variational inference algorithm | Distributions that no longer need fast exact density evaluators | Example applications |
|---|---|---|
| Importance Sampling [17] (Alg. 1, Appendix B.1) | proposal $q(x; y)$ | Nested IS [27] (Appendix B.6), Agglomerative Monte Carlo (Section 5, RAVI strategy 2), Annealed IS [29] (Appendix B.5) |
| Particle Filtering [11] (Appendix B.3) | initial proposal $q_0(x_0; y_0)$, step proposals $q_t(x_i \mid x_{t-1}, y_t)$ | Nested SMC [27] (Appendix B.6), SMC$^2$ [7] (Appendix B.7) |
| Del-Moral SMC [10] (Appendix B.3) | initial proposal $q_0(x_0)$, forward kernels $K_t(x_t \mid x_{t-1})$, reverse kernels $L_t(x_{t-1} \mid x_t)$, targets $\tilde{\pi}_t(x)$ | |
| Black-Box Variational Inference [32] (Alg. 3) | variational family $q_\theta(x; y)$ | IWAE [4] (Appendix B.2), MCVI [35] (Section 2, Appendix B.9), Variational SMC [28] (Appendix B.4) |
| Amortized Variational Inference [18] (Alg. 4) | variational family $q_\theta(x; y)$ | Amortized Rejection Sampling [26] (Appendix B.8) |
| Metropolis-Hastings (Alg. 5) | transition proposal $q(x'; x)$ | pseudo-marginal ratio MH [2] |
| Hierarchical Variational Inference [33] | variational family $q_\theta(z, x; y)$, reverse proposal $r_\theta(z; x, y)$ | Importance-Weighted HVI [38], RAVI-MCVI (Sections 2 and 5, RAVI strategy 1) |

Table 1: RAVI generalizes many algorithms for Monte Carlo and variational inference, by allowing practitioners to choose proposals, variational families, and intermediate targets for which exact density evaluators are not available. In the "example applications" column, we list both novel examples of algorithms that exploit this degree of freedom (e.g., the Agglomerative Monte Carlo algorithm we develop in Section 5), and algorithms from the literature that — as we show in Appendix B — can be viewed as instances of simpler algorithms, but with certain sophisticated proposals whose density RAVI estimates.

## 2 RECURSIVE AUXILIARY-VARIABLE INFERENCE

In this section, we introduce the RAVI framework in the context of a running example: we incorporate a chain of MCMC steps into a proposal, so that it can more accurately approximate a posterior distribution. Our approach generalizes Salimans et al. [35]'s Markov Chain Variational Inference (MCVI) algorithm, and fixes a flaw that prevents it from scaling to longer MCMC chains.

**An expressive proposal based on MCMC.** Let $p(x, y)$ be a latent-variable model and $y$ an observation. Suppose we wish to approximate $p(x \mid y)$ using an expressive proposal $q(x)$, that generates an initial location $x_0$ from a simple parametric distribution $q_0$, then iterates $M$ steps of an MCMC kernel $T$:[1]

$$q(x) = \int q_0(x_0) \left( \prod_{i=1}^{M} T(x_{i-1} \to x_i) \right) \delta_{x_M}(x) \mathrm{d}x_{0:M}.$$

Even when $q_0$ is a poor approximation to $p(x \mid y)$, $q(x)$ may

---

[1]Why incorporate $M$ MCMC steps into a proposal $q$, rather than simply running MCMC? Several reasons: (1) if we use $q$ as an importance sampling proposal, the importance weights are unbiased estimates of the marginal likelihood $p(y)$, which we can use to evaluate our model; (2) if we use $q$ as a variational family, we can optimize the ELBO to learn parameters of the initial proposal or the MCMC transition kernel; and (3) if we generate many importance sampling particles using $q$, their importance weights can in theory correct for the bias of finite-sample MCMC.

be close to the posterior, if $M$ is sufficiently high. However, because the density $q(x)$ cannot be efficiently evaluated, we cannot use $q(x)$ as a proposal within importance sampling (we have no way to evaluate the importance weight $\frac{p(x,y)}{q(x)}$), nor as a variational family in VI (we cannot estimate the ELBO $\mathcal{L} = \mathbb{E}_{x \sim q}[\log \frac{p(x,y)}{q(x)}]$ or its gradient, making it impossible to learn $p$'s or $q$'s parameters).

**Approximating proposal densities with meta-inference.** RAVI's goal is to enable inference even when we cannot compute the marginal densities of our proposals and variational families exactly. To apply RAVI, we must specify not just the proposal itself but also a *meta-inference* algorithm, bundled with the proposal into an *inference strategy*:

**Definition.** An *inference strategy* $\mathcal{S}$ *targeting* $\pi$ specifies:

- a posterior approximation $\mathcal{S}.q(x) \approx \pi(x)$[2] that either has an efficient density evaluator, or is the marginal distribution of a joint distribution with a tractable density, i.e. $\mathcal{S}.q(x) = \int \mathcal{S}.q(r, x) \mathrm{d}r$, and,
- if $\mathcal{S}.q$'s marginal density cannot be efficiently evaluated, a *meta-inference strategy* $\mathcal{S}.\mathcal{M}$, assigning to each value of $x$ an inference strategy $\mathcal{S}.\mathcal{M}(x)$ targeting $\mathcal{S}.q(r \mid x)$.

---

[2]To simplify the exposition, we assume that if an inference strategy $\mathcal{S}$ targets $\pi$, then the approximation $\mathcal{S}.q$ is *mutually absolutely continuous* with $\pi$, i.e. the measure-zero events under $\pi$ are exactly the same as those under $\mathcal{S}.q$. This requirement can be relaxed somewhat; see Appendix C.

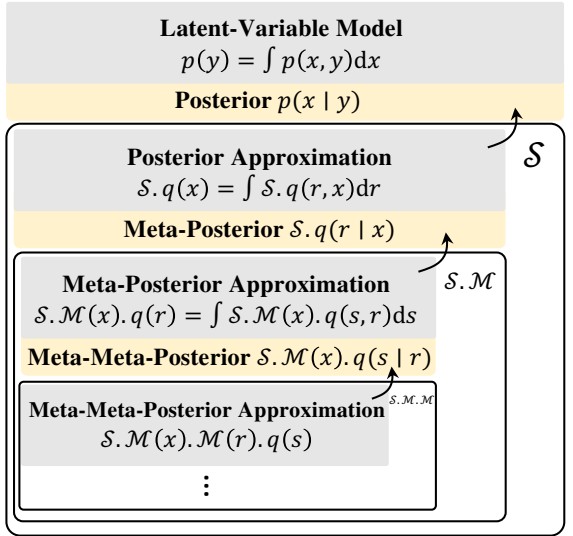

Figure 1: Structure of a RAVI inference strategy $\mathcal{S}$ targeting the posterior $p(x \mid y)$ of a latent-variable model. The proposal $\mathcal{S}.q(x) = \int \mathcal{S}.q(r, x)\mathrm{d}r$ has an intractable marginal density, so the strategy also specifies a *meta-inference* strategy $\mathcal{S}.\mathcal{M}$ that targets $\mathcal{S}.q(r \mid x)$. Nesting continues until the $q$ approximation at some layer has a tractable density, at which point no further meta-inference is needed.

Figure 1 illustrates the recursive structure of an inference strategy. The key novelty is the inclusion of *meta-inference*, in the form of *meta-posterior approximations*: additional proposals that the user specifies for inferring auxiliary variables introduced by existing proposal distributions. In our running example, we take $\mathcal{S}.q(x)$ to be our MCMC-based posterior approximation: it lacks a tractable density, but is the marginal of a tractable joint density $\mathcal{S}.q(x_{0:M}, x)$ over entire MCMC traces. A *meta-posterior approximation*, then, is a probability distribution $\mathcal{S}.\mathcal{M}(x).q(x_{0:M})$ that approximates the *meta-posterior* $\mathcal{S}.q(x_{0:M} \mid x)$: the distribution over traces of the MCMC chain, given the final location $x$.

The meta-posterior approximations enable RAVI to estimate the intractable marginal density of the top-level posterior approximation, to compute weights and gradients:

*In Monte Carlo:* If $\mathcal{S}.q(x) = \int \mathcal{S}.q(r, x)\mathrm{d}r$ is intended for use as a Monte Carlo proposal, RAVI uses meta-inference to obtain an unbiased estimate of $\frac{1}{\mathcal{S}.q(x)}$ (Algorithm 2), which is then multiplied by $p(x, y)$ to estimate the importance weight $\frac{p(x,y)}{\mathcal{S}.q(x)}$. This process relies on the *harmonic mean identity* [31], that for any meta-posterior approximation $h$,

$$\mathbb{E}_{\mathcal{S}.q(r|x)}\left[\frac{h(r)}{\mathcal{S}.q(r, x)}\right] = \frac{1}{\mathcal{S}.q(x)}\mathbb{E}\left[\frac{h(r)}{\mathcal{S}.q(r \mid x)}\right] = \frac{1}{\mathcal{S}.q(x)}.$$

(Harmonic mean estimators are infamous for having potentially infinite variance, but only when $h$ is set to a broad prior; we give a general analysis of the variance of RAVI's importance weights in Section 4.)

*In Variational Inference:* If $\mathcal{S}.q(x) = \int \mathcal{S}.q(r, x)\mathrm{d}r$ is intended as a variational family, then RAVI uses the meta-posterior approximation to formulate an *upper bound* on $\log \mathcal{S}.q(x)$: for any meta-posterior approximation $h(r)$,

$$\log \mathcal{S}.q(x) \leq \mathcal{U}(x) := \mathbb{E}_{\mathcal{S}.q(r|x)}[\log \mathcal{S}.q(r, x) - \log h(r)].$$

This follows from Jensen's inequality, and the harmonic mean identity from above. With this upper bound in hand, we formulate a surrogate ELBO $\mathcal{L}_{\mathcal{S}} = \mathbb{E}_{\mathcal{S}.q(x)}[\log p(x, y) - \mathcal{U}(x)] \leq \mathcal{L}$, which we can tractably estimate and optimize via stochastic gradient descent (Algorithm 3).

In Section 3, we show how similar estimators can be built up recursively when the meta-posterior approximations themselves have intractable marginal densities.

**A meta-inference strategy that recovers the MCVI objective [35].** In our running example, where the auxiliary randomness $r$ is a trace $x_{0:M}$ of locations visited by MCMC, one option for meta-inference is to learn neurally parameterized reverse Markov kernels $R_i(x_{i+1} \to x_i)$, and apply them in sequence to infer a plausible trace of MCMC steps leading to the final location $x$:

$$\mathcal{S}.\mathcal{M}(x).q(x_{0:M}) = \delta_x(x_M) \prod_{i=0}^{M-1} R_i(x_{i+1} \to x_i).$$

This approximation to $\mathcal{S}.q(x_{0:M} \mid x)$ has a tractable density, and so completely specifies the meta-inference strategy $\mathcal{S}.\mathcal{M}$; there is no need to specify a *meta*-meta-inference strategy. Given $\mathcal{S}$, RAVI optimizes the surrogate objective $\mathcal{L}_{\mathcal{S}} = \mathbb{E}_{x \sim \mathcal{S}.q}[\log p(x, y) - \mathcal{U}_{\mathcal{S}.\mathcal{M}(x)}]$, where

$$\mathcal{U}_{\mathcal{S}.\mathcal{M}(x)} = \mathbb{E}_{x_{0:M} \sim \mathcal{S}.q(x_{0:M}|x)}\left[\log \frac{\mathcal{S}.q(x_{0:M}, x)}{\mathcal{S}.\mathcal{M}(x).q(x_{0:M})}\right].$$

For the above choice of $\mathcal{S}.\mathcal{M}$, the RAVI objective $\mathcal{L}_{\mathcal{S}}$ exactly coincides with the Markov Chain Variational Inference (MCVI) objective of Salimans et al. [35]. In fact, RAVI unifies and generalizes many existing methods; 10 examples are collected in Appendix B.

**Analyzing MCVI within the RAVI framework.** Framing MCVI as a RAVI algorithm lets us analyze it using general theory about RAVI objectives. For example, the relative tightness of the bound $\mathcal{L}_{\mathcal{S}}$ is controlled by the quality of meta-inference:

$$\mathcal{L} - \mathcal{L}_{\mathcal{S}} = \mathbb{E}_{\mathcal{S}.q(x)}[KL(\mathcal{S}.q(x_{0:M} \mid x) \| \mathcal{S}.\mathcal{M}(x).q(x_{0:M}))].$$

We can use this characterization to analyze the MCVI objective's behavior as $M$ grows, i.e., as MCMC steps are added. Informally, as the MCMC chain begins to mix, the marginal distribution $\mathcal{S}.q(x)$ over the final location of the chain should grow closer to the posterior $p(x \mid y)$, tightening the (intractable) ELBO $\mathcal{L}$. Unfortunately, the meta-inference gap $\mathcal{L} - \mathcal{L}_{\mathcal{S}}$ *grows* with $M$, unless each kernel $R_i$ exactly

**RAVI Inference Strategy 1:** RAVI-MCVI

**Posterior Approx.** `rmcvi(M, K).q()`

    **Target of inference :** latent variable $x$
    **Auxiliary variables :** MCMC trace $x_{0:M}$

1    $x_0 \sim q_0$
2    **for** $i \in 1, \ldots, M$ **do**
3        $x_i \sim T(x_{i-1} \to \cdot)$
4    **return** $x_M$

**Meta-Posterior Approx.** `rmcvi(M, K).M(x).q()`

    **Target of inference :** MCMC trace $x_{0:M}$
    **Auxiliary variables :** SMC particles $x_{0:M}^{1:K}$, ancestor indices $a_0, a_{1:M}^{1:K}$

1    **for** $k \in 1, \ldots, K$ **do**
2        $(x_M^k, w_M^k, t_k) \leftarrow (x, q_m(x), [x])$
3    **for** $i \in M-1, \ldots, 0$ **do**
4        **for** $k \in 1, \ldots, K$ **do**
5            $a_{i+1}^k \sim \text{Discrete}(w_{i+1}^{1:K})$
6            $x_i^k \sim R_i(x_{i+1}^{a_{i+1}^k} \to \cdot)$  `// MCVI backward kernel`
7            $w_i^k \leftarrow \dfrac{q_i(x_i^k) T(x_i \to x_{i+1}^{a_{i+1}^k})}{q_{i+1}(x_{i+1}^{a_{i+1}^k}) R_i(x_{i+1}^{a_{i+1}^k} \to x_i^k)}$
8            $t_k \leftarrow [x_i^k, t_k^{a_{i+1}^k} \ldots]$
9    $a_0 \sim \text{Discrete}(w_0^{1:K})$
10   **return** $t_{a0}$

**Meta-Meta-Posterior Approx.** `rmcvi(M, K).M(x).M(x_{0:M}).q()`

    **Target of inference :** SMC particles $x_{0:M}^{1:K}$, ancestor indices $a_0, a_{1:M}^{1:K}$
    **Auxiliary variables :** None

1    **for** $i \in 0, \ldots, M$ **do**
        $b_i \sim \text{Uniform}(1, K)$
2    **for** $k \in 1, \ldots, K$ **do**
3        $(x_M^k, w_M^k) \leftarrow (x, q_m(x)])$
4    **for** $i \in M-1, \ldots, 0$ **do**
5        **for** $k \in 1, \ldots, K$ **do**
6            **if** $k = b_i$ **then**
7                $(a_{i+1}^k, x_i^k) \leftarrow (b_{i+1}, x_i)$
8            **else**
9                $a_{i+1}^k \sim \text{Discrete}(w_{i+1}^{1:K})$
10              $x_i^k \sim R_i(x_{i+1}^{a_{i+1}^k} \to \cdot)$
11            $w_i^k \leftarrow \dfrac{q_i(x_i^k) T(x_i \to x_{i+1}^{a_{i+1}^k})}{q_{i+1}(x_{i+1}^{a_{i+1}^k}) R_i(x_{i+1}^{a_{i+1}^k} \to x_i^k)}$
12   $a_0 \leftarrow b_0$
13   **return** $(a_0, a_{1:M}^{1:K}, x_{0:M}^{1:K})$

---

captures the local posterior $\mathcal{S}.q(x_i \mid x_{i+1})$. (This can be seen as an instance of the well-known *degeneracy problem* of sequential importance sampling [13, Proposition 1].) As MCMC converges, the rate of improvement in $\mathcal{L}$ slows, and the meta-inference penalty for increasing the chain's length eventually outweighs the benefit of improving the posterior approximation $\mathcal{S}.q$. The red curves in Figure 3 show this phenomenon playing out on two toy targets: we see that $\mathcal{L}_\mathcal{S}$ does become tighter as more MCMC steps are added, but only to a point, before the bound begins to *loosen*.

**Resolving the issue with improved meta-inference.** RAVI clarifies that the variational bound loosens with increasing $M$ due to poor meta-inference: as the MCMC chain grows longer, error in the learned backward kernels accumulates. This analysis also points to a solution: use a meta-inference strategy $\mathcal{S}.\mathcal{M}$ that *can* scale to longer MCMC histories.

A standard approach to resolving the degeneracy problem when inferring sequences of latent variables is *sequential Monte Carlo* (SMC) [10]. SMC tracks $K$ hypotheses about a latent sequence, periodically weighting the hypotheses and *resampling*, to clone promising particles and cull poor

ones. Using RAVI, we can use SMC for *meta-inference*: we choose $\mathcal{S}.\mathcal{M}(x).q(x_{0:M})$ to generate a collection of $K$ possible backward MCMC trajectories, using SMC, before selecting one to return. This meta-posterior approximation is shown in RAVI Inference Strategy 1.

This algorithm does not itself have a tractable marginal density: computing $\mathcal{S}.\mathcal{M}(x).q(x_{0:M})$ would require large sums over the resampling variables and intractable integrals over the particle collection. But this is where RAVI's recursive structure comes into play: a meta-inference strategy may use an intractable meta-posterior approximation, so long as we attach a *meta*-meta-inference strategy $\mathcal{S}.\mathcal{M}(x).\mathcal{M}(x_{0:M}).q(a_0, a_{1:M}^{1:K}, x_{0:M}^{1:K})$. In this case meta-meta-inference must infer the auxiliary variables of SMC (ancestor variables and unchosen trajectories), given the final chosen trajectory $x_{0:M}$. For this we can use the *conditional SMC* algorithm [1], which runs SMC, with the same auxiliary variables, but constrained to ensure that one of the $K$ particles traces the observed trajectory $x_{0:M}$. Because cSMC introduces no new auxiliary variables, it has a tractable density, and there is no need to specify a fourth layer of meta-inference. The full tower of posterior approximations is given in RAVI Inference Strategy 1.

In Section 5, we compare MCVI to `rmcvi`, for different $K$ and $M$. Figure 3 shows that meta-inference error is greatly reduced by using SMC, so that the variational bound $\mathcal{L}_\mathcal{S}$ continues to tighten as the MCMC chain grows longer.

**Using the inference strategy within a Monte Carlo algorithm, to estimate marginal likelihoods from MCMC results.** Our inference strategy $\mathcal{S}$ can also be used as proposal within Monte Carlo algorithms, such as importance sampling. In the context of our example, where $\mathcal{S}.q$ incorporates $M$ steps of a Markov chain, this allows us to assign an *importance weight* to each run of the Markov chain. The weight is an unbiased estimate of the marginal likelihood $p(y)$ of the model; thus, we can view the algorithm as a way to derive marginal likelihood estimates from MCMC runs, a task of long-standing interest in the Monte Carlo community [29]. In Section 5, we show that in some settings MCVI compares favorably a standard algorithm for the task, annealed importance sampling (AIS) [29].

## 3 ALGORITHMS

In this section, we present algorithms for using RAVI inference strategies within Monte Carlo and variational inference algorithms, as proposals and variational families.

**RAVI for Importance Sampling and SMC.** In importance sampling and SMC algorithms, proposals $q$ are used to (1) generate proposed values $x \sim q$, and (2) compute importance weights $\frac{p(x)}{q(x)}$. But in both IS and SMC, it suffices to produce *unbiased estimates* of $\frac{p(x)}{q(x)}$ [6]. RAVI exploits this

## Recursive Monte Carlo Estimation

**Algorithm 1:** RAVI Importance Sampling (IMPORTANCE)
**Input:** unnormalized target $\tilde{\pi}(x) = Z\pi(x)$
**Input:** inference strategy $\mathcal{S}$
**Output:** $(x, \hat{Z})$ properly weighted for $\pi(x)$, s.t. $\mathbb{E}[\hat{Z}] = Z$
1 **if** $\mathcal{S}.q$ *has a tractable marginal density* **then**
2     $x \sim \mathcal{S}.q$
3     $w \leftarrow \frac{1}{\mathcal{S}.q(x)}$
4 **else if** $\mathcal{S}.q(x) = \int \mathcal{S}.q(r,x)dr$ **then**
5     $(r,x) \sim \mathcal{S}.q$
6     $w \leftarrow \text{HME}(\mathcal{S}.q(\cdot \mid x), r, \mathcal{S}.\mathcal{M}(x))$
7 **return** $(x, w\tilde{\pi}(x))$

**Algorithm 2:** RAVI Harmonic Mean Estimation (HME)
**Input:** unnormalized target $\tilde{\pi}(x) = Z\pi(x)$
**Input:** exact sample $x \sim \pi$
**Input:** inference strategy $\mathcal{S}$
**Output:** unbiased estimate $\check{Z}^{-1}$ of $Z^{-1}$
1 **if** $\mathcal{S}.q$ *has a tractable marginal density* **then**
2     $w \leftarrow \mathcal{S}.q(x)$
3 **else if** $\mathcal{S}.q(x) = \int \mathcal{S}.q(r,x)dr$ **then**
4     $(r,w) \leftarrow \text{IMPORTANCE}(\mathcal{S}.q(\cdot,x), \mathcal{S}.\mathcal{M}(x))$
5 **return** $w/\tilde{\pi}(x)$

## Recursive Variational Objectives and Gradient Estimation

**Algorithm 3:** RAVI ELBO and gradient estimator (ELBO$\nabla$)
**Input:** model $p(x,y)$
**Input:** data $y$
**Input:** inference strategy $\mathcal{S}$
**Output:** unbiased estimates of $\mathcal{L}(p,y,\mathcal{S})$ and of $\nabla_\theta \mathcal{L}(p,y,\mathcal{S})$
1 **if** $\mathcal{S}.q$ *has a tractable marginal density* **then**
2     $x \sim \mathcal{S}.q$
3     $(\hat{U}, \widehat{\nabla_\theta}) \leftarrow (\log \mathcal{S}.q(x), \nabla_\theta \log \mathcal{S}.q(x) \cdot (1 + \log \mathcal{S}.q(x)))$
4     $\mathbf{g} \leftarrow \nabla_\theta \log \mathcal{S}.q(x)$
5 **else if** $\mathcal{S}.q(x) = \int \mathcal{S}.q(r,x)dr$ **then**
6     $(r,x) \sim \mathcal{S}.q$
7     $(\hat{U}, \widehat{\nabla_\theta}, \mathbf{g}) \leftarrow \text{EUBO}\nabla(\mathcal{S}.q, x, r, \mathcal{S}.\mathcal{M}(x))$
8 $\hat{L} \leftarrow \log p(x,y) - \hat{U}$
9 $\widehat{\nabla_\theta}' \leftarrow \nabla_\theta \log p(x,y) + \mathbf{g} \log p(x,y) - \widehat{\nabla_\theta}.$
10 **return** $(\hat{L}, \widehat{\nabla_\theta}')$

**Algorithm 4:** RAVI EUBO and gradient estimator (EUBO$\nabla$)
**Input:** model $p(x,y)$
**Input:** data $y$
**Input:** exact sample $x \sim p(x \mid y)$
**Input:** inference strategy $\mathcal{S}$
**Output:** unbiased estimates of $\mathcal{U}(p,y,\mathcal{S})$ and $\nabla_\theta \mathcal{U}(p,y,\mathcal{S})$
**Output:** quantity $\mathbf{g}$ (see Thm. 2)
1 **if** $\mathcal{S}.q$ *has a tractable marginal density* **then**
2     $(\hat{L}, \widehat{\nabla_\theta}) \leftarrow (\log \mathcal{S}.q(x), \nabla_\theta \log \mathcal{S}.q(x))$
3 **else if** $\mathcal{S}.q(x) = \int \mathcal{S}.q(r,x)dr$ **then**
4     $(\hat{L}, \widehat{\nabla_\theta}) \leftarrow \text{ELBO}\nabla(\mathcal{S}.q, x, \mathcal{S}.\mathcal{M}(x))$
5 $\hat{U} \leftarrow \log p(x,y) - \hat{L}$
6 $\mathbf{g} \leftarrow \nabla_\theta \log p(x,y)$
7 $\widehat{\nabla_\theta}' \leftarrow \nabla_\theta \log p(x,y) + \mathbf{g} \cdot \hat{U} - \widehat{\nabla_\theta}$
8 **return** $(\hat{U}, \widehat{\nabla_\theta}', \mathbf{g})$

## MCMC

**Algorithm 5:** RAVI Metropolis-Hastings
**Input:** model $\tilde{\pi}(x) = Z \int \pi(r,x)dr$
**Input:** proposal $q(x';x) = \int q(s,x';x)ds$
**Input:** family $\mathcal{S}(x)$ of inference strategies targeting $\pi(r \mid x)$
**Input:** family $\mathcal{M}(x,x')$ of inference strategies targeting $q(s \mid x';x)$
**Input:** initial position $x$ and estimate $\hat{Z}_x$ of $\tilde{\pi}(x)$
**Output:** next position $x'$ and estimate $\hat{Z}_{x'}$ of $\tilde{\pi}(x')$
1 $(s,x') \sim q(s,x';x)$
2 $w_{x'} \leftarrow \text{HME}(q(\cdot,x';x), s, \mathcal{M}(x,x'))$
3 $(\_, w_x) \leftarrow \text{IMPORTANCE}(q(\cdot \mid x;x'), \mathcal{M}(x',x))$
4 $(\_, \hat{Z}_{x'}) \leftarrow \text{IMPORTANCE}(\pi(\cdot \mid x'), \mathcal{S}(x'))$
5 $u \sim \text{Uniform}(0,1)$
6 **if** $u < min(1, \frac{\hat{Z}_{x'}}{\hat{Z}_x} w_{x'} w_x)$ **then**
7     **return** $(x', \hat{Z}_{x'})$
8 **else**
9     **return** $(x, \hat{Z}_x)$

degree of freedom to generate proper importance weights even when $q(x)$ is intractable. Suppose $\tilde{\pi} = Z\pi$ is an unnormalized target density, and $\mathcal{S}$ is a RAVI inference strategy targeting $\pi$. Algorithm 1 simulates $x \sim \mathcal{S}.q$ and computes an unbiased estimate $\hat{Z}$ of $\frac{\tilde{\pi}(x)}{\mathcal{S}.q(x)}$:

**Theorem 1.** *Let* $\tilde{\pi}(x) = Z\pi(x)$ *be an unnormalized target density, and* $\mathcal{S}$ *an inference strategy targeting* $\pi(x)$. *Then:*

- IMPORTANCE$(\mathcal{S}, \tilde{\pi})$ *generates* $(x, \hat{Z})$ *with* $x \sim \mathcal{S}.q$ *and* $\mathbb{E}[\hat{Z} \mid x] = Z\frac{\pi(x)}{\mathcal{S}.q(x)}$. *Furthermore, the unconditional expectation* $\mathbb{E}[\hat{Z}(\tilde{\pi}, \mathcal{S})] = Z$.

- *When* $x \sim \pi$, HME$(\mathcal{S}, x, \tilde{\pi})$ *generates* $\check{Z}$ *with* $\mathbb{E}[\check{Z}^{-1}] = Z^{-1}$.

When $\mathcal{S}.q$ has a tractable marginal density, Algorithm 1 computes an exact importance weight. Otherwise, it calls Algorithm 2, which uses the meta-inference strategy $\mathcal{S}.\mathcal{M}(x)$ to estimate $\frac{1}{\mathcal{S}.q(x)}$. The proof of Theorem 1 is by induction on the level of nesting in the strategy (see Appendix A).

**RAVI for MCMC.** When models or proposals (or both) in a Metropolis-Hastings sampler do not have tractable closed-form densities, RAVI inference strategies enable computation of MH acceptance probabilities (Algorithm 5). Intuitively, to compute the usual Metropolis-Hastings acceptance probability $\alpha = \frac{\tilde{\pi}(x')q(x;x')}{\tilde{\pi}(x)q(x';x)}$, Algorithm 5 estimates the necessary proposal densities, using HME for the forward proposal density that appears in the denominator, and IMPORTANCE for the backward proposal density that appears in the numerator. If necessary, it also uses IMPORTANCE to estimate the new model density $\tilde{\pi}(x')$.

We show the algorithm implements a stationary kernel for $\pi$ in Appendix A.5.

**RAVI for Variational Inference.** Let $p_\theta(x, y)$ be a latent-variable generative model with parameters $\theta$, and $\mathcal{S}_\theta(y)$ is a family of strategies targeting $p_\theta(x \mid y)$. Given a dataset $y$, variational inference can be applied to maximize (a lower bound on) $\log p_\theta(y)$, and also to optimize parameters of the posterior approximations in $\mathcal{S}_\theta$, to bring them closer (in KL divergence) to their targets. Let

$$\mathcal{L}(p, y, \mathcal{S}) := \mathbb{E}[\log \hat{Z}(p(\cdot, y), \mathcal{S})] \leq \log p(y)$$
$$\text{and } \mathcal{U}(p, y, \mathcal{S}) := \mathbb{E}[\log \check{Z}(p(\cdot, y), \mathcal{S})] \geq \log p(y),$$

where $\hat{Z}(\tilde{\pi}, \mathcal{S})$ is the estimate returned by IMPORTANCE (Alg. 1) on $\mathcal{S}$ and unnormalized target $\tilde{\pi}$, and $\check{Z}(\tilde{\pi}, \mathcal{S})$ is the inverse of the weight returned from HME (Alg. 2) when run with unnormalized target $\tilde{\pi}$, inference strategy $\mathcal{S}$, and an exact sample $x \sim \pi$. Because $\hat{Z}$ is an unbiased estimate of $p_\theta(y)$, and $\check{Z}^{-1}$ is an unbiased estimate of $p_\theta(y)^{-1}$, we have by Jensen's inequality that $\mathcal{L}(p, y, \mathcal{S})$ and $\mathcal{U}(p, y, \mathcal{S})$ are lower and upper bounds (respectively) on $\log p_\theta(y)$. As such, we can fit the model parameters $\theta$ to data $y$ by minimizing $\mathcal{U}(p, y, \mathcal{S})$ or maximizing $\mathcal{L}(p, y, \mathcal{S})$.

*Recursive stochastic gradient estimation.* ELBO$\nabla$ (Alg. 3) is a procedure for estimating $\mathcal{L}(p, y, \mathcal{S})$ and its gradient $\nabla_\theta \mathcal{L}(p, y, \mathcal{S})$ with respect to the parameters $\theta$ of the model and the strategy. When $(x, y) \sim p(x, y)$, EUBO$\nabla$ (Alg. 4) estimates $\mathcal{U}(p, y, \mathcal{S})$ and the gradient $\nabla_\theta \mathbb{E}_{y \sim p}[\mathcal{U}(p, y, \mathcal{S})]$. These procedures employ score function estimation of gradients, but it is straightforward to incorporate baselines within each procedure to reduce variance. Depending on $\mathcal{S}$, the reparametrization trick may also be applicable (Appendix E).

**Theorem 2.** *Given a model $p_\theta(x, y)$ and an inference strategy $\mathcal{S}_\theta$ targeting $p_\theta(x \mid y)$, Alg. 3 yields unbiased estimates of $\mathcal{L}(p, y, \mathcal{S})$ and of $\nabla_\theta \mathcal{L}(p, y, \mathcal{S})$. Furthermore, when $(x, y) \sim p_\theta$, Alg. 4 yields (i) $\hat{U}$ such that $\mathbb{E}[\hat{U} \mid y] = \mathcal{U}(p, y, \mathcal{S})$, (ii) $\widehat{\nabla_\theta}$ such that $\mathbb{E}[\widehat{\nabla_\theta}] = \nabla_\theta \mathbb{E}_{y \sim p}[\mathcal{U}(p, y, \mathcal{S})]$, and (iii) a value $\mathbf{g}$ such that for any function $R$ that does not depend on $\theta$, $\mathbb{E}[\mathbf{g} \cdot R(y)] = \nabla_\theta \mathbb{E}_{y \sim p_\theta}[R(y)]$ if $\nabla_\theta \mathbb{E}_{y \sim p_\theta}[R(y)]$ is defined.*

In Section 4, we show the tightness of the variational bounds $\mathcal{L}$ and $\mathcal{U}$ is given by sums of KL divergences between posterior approximations in $\mathcal{S}_\theta$ and their targets. Thus, optimizing these bounds improves the posterior approximations, either encouraging mass-capturing or mode-seeking behavior.

# 4 THEORETICAL ANALYSIS

We now present theorems characterizing the quality of RAVI inference: Thm. 3 concerns the variance of weights in a Monte Carlo sampler, and Thm. 4 the tightness of variational bounds. In both cases, error is related to each approximation in the RAVI strategy's divergence to its target posterior.

**Sampler variance in Monte Carlo.** Let $\tilde{\pi} = Z\pi$ be an unnormalized target density, and $\mathcal{S}$ an inference strategy targeting $\pi$. As in Section 3, we write $\hat{Z}(\tilde{\pi}, \mathcal{S})$ for the weight returned by IMPORTANCE, and $\text{Var}_{\hat{Z}}(\pi, \mathcal{S})$ for the *relative variance* of the estimator, $\text{Var}(\frac{\hat{Z}(\tilde{\pi}, \mathcal{S})}{Z})$, which does not depend on $Z$ (and therefore is a function of $\pi$, not $\tilde{\pi}$). Similarly, we write $\check{Z}(\tilde{\pi}, \mathcal{S})$ for the reciprocal of the weight returned by HME, run with an input $x \sim \pi$. $\text{Var}_{\check{Z}}(\pi, \mathcal{S})$ is its relative variance, $\text{Var}(\frac{Z}{\check{Z}(\tilde{\pi}, \mathcal{S})})$.

**Theorem 3.** *Consider an unnormalized target distribution $\tilde{\pi}(x) = Z\pi(x)$ and an inference strategy $\mathcal{S}$ targeting $\pi(x)$. Then the relative variances of the estimators $\hat{Z}(\tilde{\pi}, \mathcal{S})$ and $\check{Z}(\tilde{\pi}, \mathcal{S})$ are given by the following recursive equations:*

$$\text{Var}_{\hat{Z}}(\pi, \mathcal{S}) = \chi^2(\pi || \mathcal{S}.q) +$$
$$\mathbb{E}_{x \sim \mathcal{S}.q}\left[ \left( \frac{\pi(x)^2}{\mathcal{S}.q(x)^2} \right) \cdot \text{Var}_{\check{Z}}(\mathcal{S}.q(\cdot \mid x), \mathcal{S}.\mathcal{M}(x)) \right]$$
$$\text{Var}_{\check{Z}}(\pi, \mathcal{S}) = \chi^2(\mathcal{S}.q || \pi) +$$
$$\mathbb{E}_{x \sim \pi}\left[ \left( \frac{\mathcal{S}.q(x)^2}{\pi(x)^2} \right) \cdot \text{Var}_{\hat{Z}}(\mathcal{S}.q(\cdot \mid x), \mathcal{S}.\mathcal{M}(x)) \right]$$

*When $\mathcal{S}.q$ is tractable, the second term of each sum is 0.*

**Tightness of variational bounds.** In VI, the tightness of the variational bounds $\mathcal{L}$ and $\mathcal{U}$ can be characterized as a sum of a KL divergence and a term measuring meta-inference error. The random variables $\hat{\mathcal{L}}$ and $\hat{\mathcal{U}}$ returned by ELBO$\nabla_\theta$ and EUBO$\nabla_\theta$, respectively, are unbiased estimators of $\mathcal{L}(p, y, \mathcal{S})$ and $\mathcal{U}(p, y, \mathcal{S})$, and so can also be viewed as *biased* estimators of $\log p(y)$. Writing their bias as $\text{Bias}_{\mathcal{L}}(p, y, \mathcal{S})$ (and similarly for $\mathcal{U}$), we have:

**Theorem 4.** *Consider a joint distribution $p(x, y)$ and an inference strategy $\mathcal{S}$ targeting $p(x \mid y)$. Then the following equations give the bias of $\hat{\mathcal{L}}$ and $\hat{\mathcal{U}}$ as estimators of $\log p(y)$:*

$$\text{Bias}_{\mathcal{L}}(p, y, \mathcal{S}) = -\text{KL}(\mathcal{S}.q || p(\cdot \mid y))$$
$$- \mathbb{E}_{x \sim \mathcal{S}.q}[\text{Bias}_{\mathcal{U}}(\mathcal{S}.q, x, \mathcal{S}.\mathcal{M}(x))]$$
$$\text{Bias}_{\mathcal{U}}(p, y, \mathcal{S}) = \text{KL}(p(\cdot \mid y) || \mathcal{S}.q)$$
$$- \mathbb{E}_{x \sim p(\cdot | y)}[\text{Bias}_{\mathcal{L}}(\mathcal{S}.q, x, \mathcal{S}.\mathcal{M}(x))]$$

*where the second term in each equation is 0 when $\mathcal{S}.q$ has a tractable marginal density.*

Maximizing $\mathcal{L}$, or minimizing $\mathcal{U}$, also minimizes these KL divergences. In particular, maximizing $\mathcal{L}(p, y, \mathcal{S})$ minimizes a 'mode-seeking' KL from $\mathcal{S}.q$ to the posterior, whereas minimizing $\mathbb{E}_{y \sim p}[\mathcal{U}(p, y, \mathcal{S})]$, e.g. by following the gradients of Alg. 4, implements amortized variational inference, and encourages $\mathcal{S}.q$ to cover the mass of the posterior.

**Inference and Meta-Inference.** In both Theorems 3 and 4, the first term of the sum is a divergence between $\mathcal{S}.q(x)$, the intractable posterior approximation, and the actual target posterior $p(x \mid y)$. The other term measures the *expected*

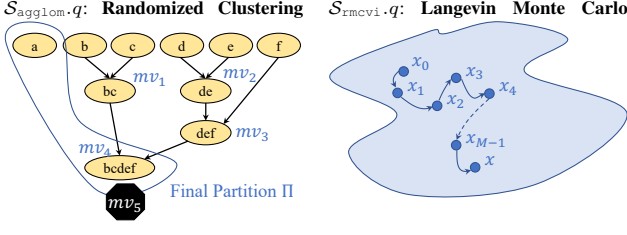

$\mathcal{S}_{\text{agglom}}.q$: **Randomized Clustering**   $\mathcal{S}_{\text{rmcvi}}.q$: **Langevin Monte Carlo**

Final Partition $\Pi$

$$\mathcal{S}_{\text{agglom}}.q(\Pi) = \sum_{mv_1} \cdots \sum_{mv_{N-|\Pi|}} \prod_{i=1}^{N-|\Pi|} q(mv_i \mid mv_{<i})$$

$$\mathcal{S}_{\text{rmcvi}}.q(x) = \int_{\mathbb{R}^M} q_0(x_0) \left( \prod_{i=1}^{M-1} q(x_i \mid x_{i-1}) \right) q(x \mid x_{M-1}) \mathrm{d}x_{0:M-1}$$

Figure 2: Illustrations of the proposals $\mathcal{S}.q$ used in each experiment. In each case, $\mathcal{S}.q$ makes a sequence of auxiliary choices before returning a final proposal (the clustering $\Pi$, or the location $x$). Sequential Monte Carlo meta-inference is used to marginalize the sequence of auxiliary variables introduced by the inference process (the merges $mv_i$ in `agglom`, and the locations $x_i$ in `rmcvi`).

| | Inference | Meta-inference | Meta-meta-inference |
|---|---|---|---|
| **agglom** | *Discrete*: $3.0 \times 10^{1928}$ | *Discrete*: $\prod_{n=|\Pi|}^{1000} \frac{n(n-1)}{2}$ | *Discrete*: $(K-1) \cdot (\prod_{n=|\Pi|}^{1000} \frac{n(n-1)}{2}) \cdot (1000-|\Pi|)(K-1)!$ |
| **rmcvi** | *Continuous*: 1 | *Continuous*: $M$ | *Continuous*: $(K-1)\cdot M$, *Discrete*: $M(K-1)!$ |

Table 2: Dimensionality of the continuous latent space, and cardinality of the discrete latent space, over which each layer's inference problem is defined. $K$ is the number of SMC particles used for meta-inference (maximum 50 for `rmcvi`, 5 for `agglom`). In `rmcvi`, $M$ is the number of MCMC steps (maximum 100 in our experiments).

quality of meta-inference. Thus the overall error of a RAVI algorithm can be understood as decomposing cleanly into (1) the mismatch between the posterior and the intractable proposal, and (2) the error introduced by meta-inference.

# 5   EXPERIMENTS

## 5.1   IMPROVING MCVI

In Section 2, we developed a variant of Salimans et al. [35]'s MCVI algorithm that used SMC for meta-inference. In Figure 3, we compare vanilla MCVI to the RAVI variant, with varying $K$ (number of particles used for meta-inference) and $M$ (number of MCMC steps in the variational family).

**Experimental details.**[3] For the MCMC kernel $T$, we use Langevin ascent with step size $0.015$. For the

---

[3]Code is available: https://github.com/probcomp/ravi-uai-2022

meta-inference proposals $R_i(x_{i+1} \rightarrow x_i)$, we use $\mathcal{N}(x_i; f_\mu(x_{i+1}, i), e^{f_{\log\sigma}(x_{i+1}, i)})$, where $f$ is a 4-layer MLP, the step number $i$ is encoded as a one-hot vector, and $f$ outputs the mean $\mu$ and log standard deviation $\log\sigma$ for a conditionally Gaussian proposal. The same $f$ is used for each experiment, and is trained on forward rollouts of MCMC (equivalent to using Alg. 3 on `rmcvi` with $K = 1$). The unimodal model is Gaussian with $\sigma = 0.2$, and the multimodal model is a mixture of 3 Gaussians with standard deviations $0.2, 0.3$, and $2.0$. The distributions $q_i$ used for importance weighting during sequential Monte Carlo meta-inference are Gaussians with learned $\mu$ and $\sigma$.

**Results.** Figure 3 plots the gap $\log p(y) - \mathcal{L}$ for each algorithm's variational bound $\mathcal{L}$. By Theorem 4 this gap is the sum of two terms: $KL(\mathcal{S}.q||p(x \mid y))$ and the expected meta-inference divergence $\mathbb{E}_{x\sim\mathcal{S}.q}[KL(\mathcal{S}.q(x_{0:M} \mid x)||\mathcal{S}.\mathcal{M}(x).q(x_{0:M}))]$. The first term is constant across the algorithms, since they all use the same MCMC-based posterior approximation, so the plots primarily illustrate differences in the quality of meta-inference. MCVI's meta-inference steadily worsens as the chain's length grows, and after 15-25 steps, the meta-inference cost of adding new steps outweighs the benefits to $\mathcal{S}_{\text{mcvi}}.q$, causing the bound $\mathcal{L}$ to loosen. Our variant, with SMC meta-inference, does not suffer the same penalty, and continues to improve as more steps are added. As discussed in Section 2, the same inference strategy (`rmcvi`) can be used within an importance sampler to derive unbiased marginal likelihood estimates from MCMC runs. The right-hand plot in Figure 3 shows that this technique can yield accurate estimates with less computation than AIS [29], at least on simple targets. (To fairly account for the computational cost of meta-inference, in the RAVI algorithm we multiply $M$ by $K$ when plotting the total number of MCMC steps.) Because the variance of AIS is bounded below by sums of divergences between subsequent pairs of intermediate target distributions, the MCMC chain must be long enough to support a very fine annealing schedule, without large jumps. By contrast, RAVI-MCVI requires only that the marginal distribution of the chain be a good approximation to the posterior, and that SMC meta-inference is sufficiently accurate. For some problems, this may be less expensive than the long chain required by AIS.

## 5.2   AGGLOMERATIVE CLUSTERING FOR DIRICHLET PROCESS MIXTURES

A promising application of RAVI is to transform heuristic randomized algorithms into unbiased and consistent Monte Carlo estimators, by using them as proposal distributions. In this section, we design a RAVI inference strategy for clustering in Dirichlet process mixtures, based on a randomized agglomerative clustering algorithm (Inference Strategy 2).

**Datasets and Models.** We test our algorithm on three clus-

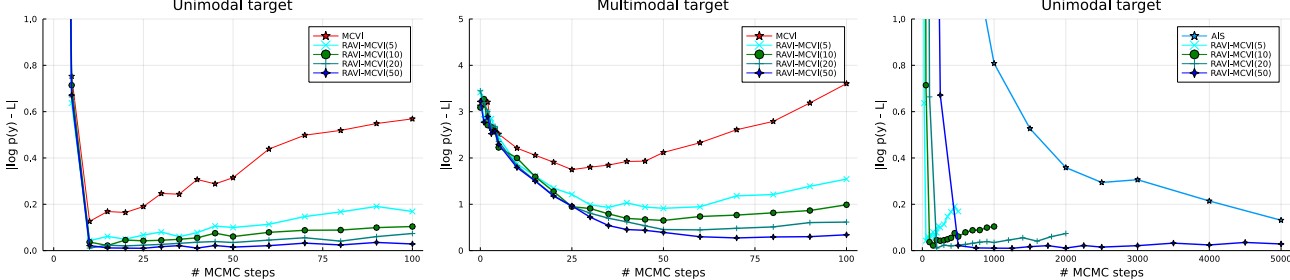

Figure 3: Improving Markov Chain Variational Inference with RAVI. **Left and Middle:** On unimodal and multimodal targets, MCVI begins to degrade after 15-25 steps of MCMC. RAVI-MCVI with sufficiently many particles continues to improve as more MCMC steps are added. **Right:** When MCMC converges quickly to a reasonable approximation of the posterior, RAVI-MCVI can give more accurate estimates of marginal likelihoods than standard techniques such as AIS. The $x$ axis of this plot counts total MCMC steps simulated, whether as part of inference or meta-inference; for RAVI-MCVI($K$), this is $KM$, where $M$ is the length of the forward Markov chain and $K$ is the number of SMC particles used for meta-inference.

**RAVI Inference Strategy 2:** Agglomerative Clustering
**Posterior Approx.** `agglom(X,K).q()`
    **Target of inference :** partition $\Pi$ of dataset $X$
    **Auxiliary variables :** merge sequence $mv_{1:|X|-|\Pi|}$
1    $\Pi \leftarrow \{\{x\} \mid x \in X\}$      // Initial partition
2    **for** $l \in 1, \ldots, |X|$ **do**
3        **for** *unordered pair* $\{i,j\}$ *of clusters in* $\Pi$ **do**
4            $w_{\{i,j\}} \leftarrow \pi((\Pi \setminus \{i,j\}) \cup \{i \cup j\})$
5        $w_{\text{stop}} = \pi(\Pi)$
6        $All \leftarrow \{\text{stop}\} \cup \{\{i,j\} \mid i,j \in \Pi\}$
7        $mv_l \sim \text{Discrete}(\{m \Rightarrow w_m \mid m \in All\})$
8        **if** $mv_l = \text{stop}$ **then break**
9        $\Pi \leftarrow (\Pi \setminus mv_l) \cup (\cup mv_l)$     // Perform the merge
10   **return** $\Pi$

**Meta-Posterior Approx.** `agglom(X,K).M(Π).q()`
    **Target of inference :** merge sequence $mv_{1:|X|-|\Pi|}$
    **Auxiliary variables :** particles $mv_{1:|X|-|\Pi|}^{1:K}$, ancestors $a_{1:|X|-|\Pi|}^{1:K}$
1    **for** $k \in 1, \ldots, K$ **do** $\Pi_0^k, tr_k \leftarrow \{\{x\} \mid x \in X\}, []$
2    **for** $l \in 1, \ldots, |X| - |\Pi|$ **do**
3        **for** $k \in 1, \ldots, K$ **do**
4            **for** *unordered pair* $\{i,j\}$ *in* $\Pi_{l-1}^k$ **do**
5                $w_{\{i,j\}} \leftarrow \pi((\Pi_{l-1}^k \setminus \{i,j\}) \cup \{i \cup j\})$
6            $w_{\text{stop}} = \pi(\Pi_{l-1}^k)$
7            $All \leftarrow \{\text{stop}\} \cup \{\{i,j\} \mid i,j \in \Pi_{l-1}^k\}$
8            $Ok \leftarrow \{\{i,j\} \in All \mid \exists c \in \Pi . i \cup j \subseteq c\}$
9            $mv_l \sim \text{Discrete}(\{m \Rightarrow w_m \mid m \in Ok\})$
10          $\Pi_{l-1}^k \leftarrow (\Pi_{l-1}^k \setminus mv_l) \cup (\cup mv_l)$
11          $tr_k \leftarrow [tr_k \ldots, mv_l]$
12          $W_l^k \leftarrow \frac{\sum_{m \in Ok} w_m}{\sum_{m \in All} w_m}$
13        **for** $k \in 1, \ldots, K$ **do**
14          $a_l^k \sim \text{Discrete}(W_l^{1:K})$     // resampling step
15          $\Pi_l^k, tr_k \leftarrow \Pi_{l-1}^{a_l^k}, tr_{a_l^k}$
16   **return** $[tr_1 \ldots, \text{stop}]$

**Meta-Meta-Posterior** `agglom(X,K).M(Π).M(mv_1:|X|-|Π|).q()`
    // Conditional SMC (omitted for space, but
        similar to that of rmcvi)

tering problems. The first is a synthetic 1D dataset sampled from a Dirichlet process (DP) mixture prior. The second is a standard benchmark dataset of galaxy velocities [14, 8], which we model using a DP mixture with Gaussian likelihoods and $\alpha = 1$. The last is a data-cleaning task, correcting typos in 1k strings from Medicare records [23]. We adapt the generative model of Lew et al. [19]. Using an English character-level bigram model $H(s) = h(s_1) \prod_{i=2}^{|s|} h(s_i \mid s_{i-1})$, we model the data $\{y_i\}$ with a DP prior:

$$G \sim DP(H, \alpha = 1.0), \quad x_i \mid G \sim G, \quad y_i \mid x_i \sim f(\cdot \mid x_i)$$

Here, the likelihood $f(y_i \mid x_i)$ models typos. We set $f$ to be

$$f(y_i \mid x_i) \propto \begin{cases} \mathbf{1}[x_i = y_i] & (x_i, y_i) \notin \mathcal{L} \times \mathcal{L} \\ \frac{\text{NegBin}(\tau(x_i,y_i);\lceil \frac{|s|}{5}\rceil, 0.9)}{(5.09|s|)^{\tau(x_i,y_i)}} & (x_i, y_i) \in \mathcal{L} \times \mathcal{L} \end{cases},$$

where $\tau(x_i, y_i)$ is the Damerau-Levenshtein edit distance between $x_i$ and $y_i$, and $\mathcal{L}$ is the set of all observed strings $\{y \mid \exists i . y = y_i\}$.[4] We perform inference in a collapsed version of the model, with the $x_i$ marginalized out:

$$\Pi \sim CRP(n = N, \alpha = 1.0)$$
$$y_I \mid \Pi \sim F(y_I).$$

Here $\Pi$ is a partition, $I$ ranges over the components of $\Pi$ (each of which is a subset of data indices), and $F(y_I) = \sum_{x \in \mathcal{L}} h(x) \prod_{i \in I} f(y_i \mid x)$ is the marginal likelihood of $y_I$ as a sequence of noisy observations of a latent string.

**Baseline.** We compare to an SMC baseline, inspired by PClean's inference [19], that targets a sequence of posteriors, where the $t^{\text{th}}$ posterior incorporates the first $t$ datapoints. The SMC proposal is locally optimal, assigning the newest datapoint to an existing component $I$ with probability proportional to $\frac{|I|}{t+\alpha-1} \cdot F(y_I \cup \{y_t\})$, or to a new component with probability proportional to $\frac{\alpha}{t+\alpha-1} \cdot F(\{y_t\})$. We do not compare to a Gibbs sampling baseline, as Gibbs sampling does not yield marginal likelihood estimates, but do perform a Gibbs rejuvenation sweep every 20 iterations of SMC.

**RAVI algorithm.** We apply Algorithm 1 to the inference strategy `agglom` (Inference Strategy 2). The strategy is based on a randomized agglomerative clustering algorithm: each datapoint begins in its own cluster (L1), and we repeatedly choose to either merge two clusters (L9) or stop and propose the current partition (L8). The sequence of merge decisions $mv_1, \ldots, mv_{|X|-|\Pi|}$ are the auxiliary variables of

---

[4]We assume that the data $\mathcal{L}$ includes at least one example of every clean string. When $x_i \in \mathcal{L}$, we model a negative-binomially distributed number of typos, where the number of trials depends on the length of the string.

| | $\hat{\mathcal{L}}$ |
|---|---|
| Gaussian likelihood [8], synthetic data | |
|    SMC + adapted proposals | $-125.09 \pm 0.38$ |
|    RAVI agglomerative clustering | $-125.97 \pm 1.62$ |
| Gaussian likelihood [8], Galaxy data [14] | |
|    SMC + adapted proposals | $-426.20 \pm 1.26$ |
|    RAVI agglomerative clustering | $\mathbf{-423.03 \pm 0.94}$ |
| PClean typos likelihood [19], Hospital data [23] | |
|    SMC + adpated proposals | $-40,239 \pm 1,532$ |
|    RAVI agglomerative clustering | $\mathbf{-13,851.0 \pm 0.01}$ |

Table 3: RAVI agglomerative clustering vs. SMC baseline.

our proposal distribution; the final output is the proposed clustering $\Pi$. Our meta-inference $\texttt{agglom}(X, K).\mathcal{M}(\Pi).q$ infers the sequence of merges from the observed clustering $\Pi$, using $K$-particle SMC with proposals that mimic the forward process but choose only from a restricted set $Ok$ of possible merges (L8), to avoid making any choices that disagree with $\Pi$. SMC introduces additional auxiliary variables, so we also include a conditional SMC meta-meta-posterior approximation (not shown, but nearly identical to `rmcvi`'s).

**Results.** Table 3 shows average log marginal likelihood estimates; higher is better. On synthetic Gaussian data, the algorithms perform comparably. On the galaxy data, RAVI agglomerative clustering finds modes that SMC misses, leading to a 3-nat improvement in the average log marginal likelihood. In the Medicare data example, SMC misses the ground-truth clustering and hypothesizes many unlikely typos to explain the data. The RAVI agglomerative clustering is less greedy, considering $O(N^2)$ possible merges at each step, rather than $O(N)$. As such, it is able to find the ground truth clustering, correctly identifying all typos (unlike PClean [19], which achieves only 90% accuracy on this dataset) and reporting a log marginal likelihood thousands of nats higher than the SMC algorithm.

# 6  RELATED WORK AND DISCUSSION

**Related work.** RAVI builds on and generalizes recent work from both the Monte Carlo and variational inference literatures. For example, Salimans et al. [35] and Ranganath et al. [33] showed how auxiliary variables could be used to construct and optimize variational bounds for specific families of expressive variational approximations. Sobolev and Vetrov [38] presented tighter bounds in a more general setting. RAVI is a further generalization, in two directions: first, we show that these bounds arise from particular choices of meta-inference strategy, and can be tightened by improving meta-inference; and second, we extend the results to the Monte Carlo setting, enabling learned variational families to be used as IS, SMC, or MH proposals. We also provide general theorems about the variance of RAVI samplers and the bias of RAVI variational bounds, which can be applied to analyze both new and existing algorithms.

RAVI is also related to other compositional or unifying frameworks for thinking about broad classes of inference algorithms [21, 44, 37, 36, 22, 9, 39, 30, 43, 3, 40, 15, 16], some of which involve recursive constructions [27, 10, 12].

However, to our knowledge, RAVI's inference strategies are novel. For example, although RAVI and Nested IS (NIS) [27] are both approaches to inference with 'intractable proposals,' NIS *approximately samples* a proposal distribution with a *tractable* (unnormalized) density, whereas RAVI *approximates the density* of a proposal that *can* be simulated tractably, but whose marginal density (even unnormalized) is intractable. As another example, Domke and Sheldon [12]'s framework of *estimator-coupling pairs* constructs variational bounds and marginal likelihood estimators recursively, but unlike in RAVI, the posterior approximations cannot be used to formulate objectives for *amortized* VI or as components of Metropolis-Hastings proposals.

Finally, researchers have used meta-inference to construct bounds on KL divergences [8] and other information-theoretic quantities [34]. In Appendix D, we show how to apply such bounds in the general RAVI setting.

**Outlook and Limitations.** RAVI expands the design space for Monte Carlo and variational inference. It gives unifying correctness proofs for over a dozen methods from the literature, and novel theorems that characterize their behavior. Experiments show that RAVI helps to design algorithms that significantly improve accuracy over previously introduced Monte Carlo and variational inference methods. However, some difficulties remain. For example, the gradient estimators we present (Algs. 3 and 4) have high variance for some strategies $\mathcal{S}$; in Appendix E, we give estimators that exploit the reparameterization trick, but they only help when the proposals in $\mathcal{S}$ can be reparameterized, which is not the case, e.g., for SMC. In these cases, RAVI can still be used to derive objectives for optimization, but practitioners will need other ways of reducing the variance of gradient estimates; many results from the literature [24, 41] should apply.

Another difficulty is that RAVI algorithms can be complex to implement. We are exploring an automated implementation based on probabilistic programming languages [9, 42]: if the posterior and meta-posterior approximations in a RAVI strategy $\mathcal{S}$ are given as probabilistic programs, we can provide Algs. 1-5 as higher-order functions, which automate the necessary densities, gradients, and MCMC acceptance probabilities. This could be viewed as a generalization of existing PPL support for *programmable inference* [21, 22, 9, 20].

**Acknowledgements**

The authors are grateful to Feras Saad, Tan Zhi-Xuan, Ben Sherman, Cameron Freer, George Matheos, Sam Witty, McCoy Becker, Jan-Willem van de Meent, Sam Stites, Eli Sennesh, Cathy Wong, and Nishad Gothoskar for useful conversations and feedback, and to our anonymous referees for helpful feedback on earlier drafts of the paper. This material is based on work supported by the NSF Graduate Research Fellowship under Grant No. 1745302.

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
