# OpenReview forum: "Recursive Monte Carlo and Variational Inference with Auxiliary Variables"
_auai.org/UAI/2022/Conference — UAI 2022 Poster_

### Official Review · Reviewer_9HFJ · 2022-04-09

**Q2(1) Originality/Novelty:** 3
**Q2(2) Significance/Impact:** 3
**Q2(3) Correctness/Technical Quality:** 3
**Q2(6) Clarity Of Writing:** 3
**Q6 Overall Score:** 6
**Q8 Confidence In Your Score:** 3

**Q1 Summary And Contributions:**

The paper proposes a meta learning based algorithm  to obtain intractable proposal densities in order to make proposals more similar to the posterior, but this comes at a cost since those densities cannot be evaluated.  Moreover, the authors seek to show that some of the existing prior work can be seen as special cases of their method for obtaining highly expressive approximating densities.

**Q2 Assessment Of The Paper:**

More detailed information regarding each of these aspects is given below:

**Q2(4) Quality Of Experiments (Optional):**

3: Good: The experimental evaluation is adequate, and the results convincingly support the main claims.

**Q2(5) Reproducibility:**

4: Excellent: Key resources (e.g., proofs, code, data) are available and key details (e.g., proof sketches, experimental setup) are comprehensively described for competent researchers to confidently and easily reproduce the main results.

**Q3 Main Strengths:**

- Introduction is well written.
-Prior work is sufficiently covered.
- I also liked the explanatory style of the paper.
- A lot of MCVI algorithms have been talked about and how they relate to RAVI
-The paper explains its limitations well.
- Theorem 3 and 4 seem useful and are well explained


**Q4 Main Weakness:**



Some comments and questions for the authors(not necessarily weaknesses):
- I have a general question for all these MCVI approaches,
this paper explains some problems with the meta inference approach that the error accumulates with larger values of M and bound becomes looser, and then introduces an elaborate and complex set of ideas from Sequential MC, this begs the questions , can a practitioner not just use MCMC in place of this complex algorithms.
- How does value of M change with dimensions or with the tail thickness of proposal and target.
- What is the time complexity for these algorithms compared to MCMC/HMC.
- No high dimensional expt. as far as I can understand
- How loose are the two bounds from each other ?
- The variances in Th3 use chi-square divergences, atleast one of those will be infinite when the target is heavy tailed(t-density) and proposal light tailed(Gaussian) or the other way round.

**Q5 Detailed Comments To The Authors:**

Some comments:
- Introduction can improve if you can cite an example where the proposal marginal density q(x) can be evaluated and when it cannot be, and in what applications This will help motivate the reader.
- Fig2, what are the unimodal and multimodal targets ?(I see it much later in Expts. , maybe can be done earlier)
- Confusing notation, difference between \hat{\nabla_{\theta}} and \hat{\nabla_{\theta}}', are those quantities the same for gradients of ELBO and EUBO. Theorem 2 talks about  \hat{\nabla_{\theta}}  but not  \hat{\nabla_{\theta}}', (it is probably the unbiased estimate of ELBO and EUBO both), I found it confusing and probably unintuitive that it is the same symbol for unbiased estimate of  gradient for both ELBO and EUBO, if that is the case.
- Theorem 2 can maybe be broken into smaller theorems to improve readibility.





**Q7 Justification For Your Score:**

The paper is mostly solid with theory and generalises many existing methods under one umbrella, I would like to get some comments from the authors about the questions I have asked(which are hopefully constructive) before revising my score.

**Q9 Complying With Reviewing Instructions:**

1: Yes.

---

### Official Review · Reviewer_q4E9 · 2022-04-11

**Q2(1) Originality/Novelty:** 3
**Q2(2) Significance/Impact:** 3
**Q2(3) Correctness/Technical Quality:** 3
**Q2(6) Clarity Of Writing:** 3
**Q6 Overall Score:** 6
**Q8 Confidence In Your Score:** 3

**Q1 Summary And Contributions:**

The paper presents the RAVI framework, which includes new algorithms for IS, VI, SMC, and MH using proposals that
do not have tractable marginal densities.  The method is applied to construct a novel variant of Salimans et al and it shows improved performance.

**Q2 Assessment Of The Paper:**

More detailed information regarding each of these aspects is given below:

**Q2(4) Quality Of Experiments (Optional):**

3: Good: The experimental evaluation is adequate, and the results convincingly support the main claims.

**Q2(5) Reproducibility:**

2: Fair: Key resources (e.g., proofs, code, data) are unavailable but key details (e.g., proof sketches, experimental setup) are sufficiently well-described for an expert to confidently reproduce the main results.

**Q3 Main Strengths:**

The is a well written paper presenting a unifying inference framework. The paper is backed up with several theorems that characterize the impact of the approximations used by RAVI on overall inference quality. The extension of Salimans et al auxiliary VI-MCMC method  is interesting and it shows improved performance in the experiments.

**Q4 Main Weakness:**

Not sure if this so called meta-inference procedure is really novel.  For example, the bound in the variational inference at the end of page 2
looks similar to the Jaakola's bound http://www.dccia.ua.es/~domingo/robots/mixture-mean-field.pdf


**Q5 Detailed Comments To The Authors:**

The paper tries to unify many different inference algorithms using the RAVI framework. However, I do think RAVI can have very high variance so that it can become impractical in high dimensional problems. It would be useful if you discuss this further together with potential solutions.

**Q7 Justification For Your Score:**

Well written paper presenting an interesting unifying inference framework.

**Q9 Complying With Reviewing Instructions:**

1: Yes.

---

### Official Review · Reviewer_2dVN · 2022-04-13

**Q2(1) Originality/Novelty:** 3
**Q2(2) Significance/Impact:** 2
**Q2(3) Correctness/Technical Quality:** 3
**Q2(6) Clarity Of Writing:** 2
**Q6 Overall Score:** 5
**Q8 Confidence In Your Score:** 3

**Q1 Summary And Contributions:**

The paper under consideration is devoted to the iterative scheme of auxiliary variable inference for Bayesian analysis via MCMC and VI.
In the Bayesian analysis independently of used methods (MCMC or VI) appears the problem of intractable densities. In the current manuscript, the authors try to tackle this issue by the general scheme called recursive auxiliary variable inference (RAVI).  It is shown that in quite a general scenario RAVI is accurate.

**Q2 Assessment Of The Paper:**

More detailed information regarding each of these aspects is given below:

**Q2(4) Quality Of Experiments (Optional):**

3: Good: The experimental evaluation is adequate, and the results convincingly support the main claims.

**Q2(5) Reproducibility:**

3: Good: Key resources (e.g., proofs, code, data) are available and key details (e.g., proofs, experimental setup) are sufficiently well-described for competent researchers to confidently reproduce the main results.

**Q3 Main Strengths:**

- Introducing the general framework for Bayesian inference  ML.
- Generic strategy suitable for both MCMC and variational inference
- Rather convincing simulation details.

**Q4 Main Weakness:**

- In general  VI and MCMC inference require different properties of densities to work well, it is hard to believe that one general framework could be effective for both.
 -The details of constructing auxiliary variables in particular cases are not clear, at last for me.
- I do not find theoretical guarantees that the iterative procedure will end on tractable density
- The notation is difficult to follow, the number of nested approximations is not included in the notation. All densities are denoted by the same letter q and approximation by S

**Q5 Detailed Comments To The Authors:**

The paper is written on a very high abstract level. I would like to see some toy examples to get a better intuition of how RAVI  works.

**Q7 Justification For Your Score:**

The paper presents a very general framework for Bayesian inference. I believe that presentation of results should be improved.

**Q9 Complying With Reviewing Instructions:**

1: Yes.

---

### Official Review · Reviewer_w6YF · 2022-04-13

**Q2(1) Originality/Novelty:** 2
**Q2(2) Significance/Impact:** 3
**Q2(3) Correctness/Technical Quality:** 3
**Q2(6) Clarity Of Writing:** 2
**Q6 Overall Score:** 5
**Q8 Confidence In Your Score:** 3

**Q1 Summary And Contributions:**

This paper proposes a framework for fitting approximations to intractable unnormalized distributions (motivated by posterior inference) by approximately marginalizing out auxiliary variables. This strategy is applied recursively in situations where it is difficult to marginalize out the auxiliary variables accurately. This framework describes a large number of approaches that have been proposed over the years.

**Q2 Assessment Of The Paper:**

More detailed information regarding each of these aspects is given below:

**Q2(4) Quality Of Experiments (Optional):**

3: Good: The experimental evaluation is adequate, and the results convincingly support the main claims.

**Q2(5) Reproducibility:**

3: Good: Key resources (e.g., proofs, code, data) are available and key details (e.g., proofs, experimental setup) are sufficiently well-described for competent researchers to confidently reproduce the main results.

**Q3 Main Strengths:**

The proposed framework is quite general, and offers a grammar for composing inference algorithms that applies to essentially all modern variational inference. Details are given for how to derive many existing algorithms in this framework, and the derivations are generally pretty clear. Examples are given of applying this sort of composition to improve the MCVI algorithm of Salimans et al. and to derive new inference algorithms for data-cleaning and clustering models.

**Q4 Main Weakness:**

I'm not sure how high the level of novelty here is. (This is often an issue with generalizing frameworks, where clearly the generalizing grammar has been "in the air" for a while but perhaps not clearly articulated.)

The main issue is that the non-recursive version of the scheme seems equivalent to a straightforward SMC-within-VI meta-strategy (see detailed comments). That said, the idea of recursing on this scheme is interesting, and the observation that a huge number of strategies can be described under this rubric is also a nice contribution.

I also don't love the "S.M(x).M(r).q(s)" notation. It seems like there's a single inference model, which could just be called q(s, r, x), and a single "meta" model, which could just be called p(s, r, x, y).

**Q5 Detailed Comments To The Authors:**

The non-recursive proposal seems equivalent to the following straightforward application of the now well-known strategy of SMC within VI:
1. Construct a joint distribution q(r, x) = q(r)q(x | r) such that the marginal q(x) is hopefully a decent approximation to p(x | y).
2. Construct an auxiliary reverse distribution p(r | x) to approximate q(r | x).
3. Tune any parameters of q(r, x) and p(r | x) to maximize
E_q[log p(r, x, y) - log q(r, x)] = log p(y) - KL(q(r, x)||p(r, x | y) ≤ log p(y).

In the paper's notation, p(r | x) is S.M(x).q(r). I think this notation obscures the fact that this is just the usual SMC/importance-sampling strategy applied with auxiliary variables r.

The novelty in the paper is iterating this scheme to deal with the situation where the density q(r) is itself the intractable marginalization of some distribution q(s, r), yielding the bound
E_q[log p(s, r, x, y) - log q(s, r, x)] = log p(y) - KL(q(s, r, x)||p(s, r, x | y) ≤ log p(y).
Again, this is just the usual importance-sampling-within-VI stuff, but this sort of recursive application of it doesn't seem to be widespread.



Notation nit: for the equation at the bottom right of page 2, there should probably be parentheses around the product; otherwise you’re technically raising δ_{x_M} and dx_{0:M} to the Mth power.

**Q7 Justification For Your Score:**

As I said above, I found it difficult to convince myself that the broadest contributions of this paper were really all that novel, and I think the notation and terminology obscure the ways in which this framework should be extremely straightforward to derive.

But I do see potential value in this kind of work, and could probably be convinced to bump up my score if there's something I've missed.

**Q9 Complying With Reviewing Instructions:**

1: Yes.

---

### Decision · Program_Chairs · 2022-05-15

**Decision:**

Accept (Poster)

**Comment:**

Meta Review: Note: one reviewer wanted to raise their score from 5 to 6, but was not able to do it in the system after the rebuttal phase.

The paper proposes an intriguing framework for recursive approximate inference, meaning that intractable variational distributions are recursively approximated via 'meta-inference'.

Pros:
Interesting framework subsuming many approximate inference techniques. Convincing experimental evaluation. Satisfying theoretical treatment.

Cons:
Novelty and originality not entirely clear. It seems that the proposed framework mainly "digests" various streams of research in one monolithic framework (which however, has value on its own right).

quality: good-very good
clarity: good
originality: fair-good
significance: good